# Tibial Plateau Leveling Osteotomy following Tibial Tuberosity Advancement Cage Removal: A Case Report

**DOI:** 10.3390/ani13223444

**Published:** 2023-11-08

**Authors:** Yauheni Zhalniarovich, Marta Mieszkowska, Magdalena Morawska-Kozłowska

**Affiliations:** Department of Surgery and Radiology, Faculty of Veterinary Medicine, University of Warmia and Mazury in Olsztyn, Oczapowskiego 14, 10-718 Olsztyn, Poland; marta.mieszkowska@uwm.edu.pl (M.M.); magdalena.morawska@uwm.edu.pl (M.M.-K.)

**Keywords:** cranial cruciate ligament, stifle, osteotomy, TPLO, TTA, complications

## Abstract

**Simple Summary:**

Persistent knee joint instability has been recognized as a potential postoperative consequence after Tibial Tuberosity Advancement (TTA) causing chronic lameness. A six-year-old male Labrador retriever (38 kg) with persistent lameness lasting six months after TTA Rapid surgery has been examined. During orthopedic examination, the lameness was subjectively graded 3/5 and the positive drawer and tibial compression tests were performed. Radiographical measurements were performed; the preoperative tibial plateau angle was 27° and the patella ligament angle relative to the tibial plateau was 102°. The decision was made to completely remove the TTA Rapid cage and all screws to have enough room to perform a TPLO and apply a 3.5 LCP plate. Long-term follow-up showed radiologically excellent healing with fusion of the gap and disappearance of the osteotomy line. Six months postoperatively, no lameness was detected at a walk and trot. The owner was completely satisfied and reported the dog being free of lameness even after long walks.

**Abstract:**

The purpose of this case report is to describe the functional and clinical outcome of a tibial plateau leveling osteotomy (TPLO) in a dog with joint instability and persistent lameness following a Tibial Tuberosity Advancement surgery (TTA) Rapid. A six-year-old male Labrador retriever (38 kg) with a tibial plateau angle of 27° and a patella ligament to tibial plateau angle of 102° and persistent lameness lasting six months after TTA Rapid surgery has been examined. During orthopedic examination, the lameness was subjectively graded 3/5 and the positive drawer and tibial compression tests were performed. The TTA Rapid cage and all screws were completely removed from the tibia to have enough room to perform a TPLO radial cut. A lameness score evaluation, client satisfaction and radiographic follow-up were performed at 4 weeks, 8 weeks and 6 months postoperatively. Long-term follow-up showed radiologically excellent healing with fusion of the gap and disappearance of the osteotomy line. Six months postoperatively, no lameness was detected at a walk and trot. The owner was completely satisfied and reported the dog being free from lameness even after long walks. No complications related to the TPLO surgery occurred.

## 1. Introduction

Cranial cruciate ligament (CCL) rupture is the most common orthopedic condition affecting the hind limb in dogs [1]. Several surgical techniques have been developed in order to prevent the cranial tibial thrust, performing either dynamic or static stabilization [2,3,4]. Tibial Tuberosity Advancement (TTA) and tibial plateau leveling osteotomy (TPLO) are the most commonly used techniques for dynamic stabilization. The goal of these techniques is to neutralize the cranial tibial thrust and thus achieve knee joint stability during weight bearing [5,6,7]. A few comparative studies have investigated the long-term outcome following TTA and TPLO and both have been reported to be associated with excellent outcomes [8,9,10]. Some comparative studies showed that TPLO seemed to be superior to TTA in long-term functional outcomes [11,12]. However, the excellence of one surgical technique over another has not been determined. A recent prospective study using the kinetic platform demonstrated that TPLO and TTA were equally effective in promoting the recovery of weight bearing over the short-term postoperative period in dogs [13]. An in vivo experimental study on beagle dogs reported that lameness scores were not different between TTA and TPLO limbs, but the meniscal total pathology score was significantly higher in TTA than in TPLO stifles. Medial meniscal tears occurred in 6/10 TTA stifle joints and 0/10 TPLO stifles at 12 weeks postoperatively and in 5/5 TTA stifles and 1/5 TPLO stifles at 7 months postoperatively [14].

The surgical procedures are associated with a wide range of postoperative complications, ranging in intensity from mild to catastrophic [11,14]. Some postoperative complications such as surgical site infection (SSI) could arise after any surgical procedure. Others are specific to these procedures, such as tibial tuberosity fractures, implant loosening or tibial fractures after TTA and TPLO [14]

Prolonged knee joint instability has been recognized as a possible postoperative consequence after TTA and TPLO [15,16]. It is also reported that cranio-caudal stifle instability during walking was found even when the patellar tendon angle was sufficiently corrected or even overcorrected after TTA procedures [15]. To the best of the authors’ knowledge, the use of TPLO surgery to treat chronic lameness and instability following TTA Rapid with implant removal is unreported. The secondary goal is to deliver a long-term functional and clinical outcome of the TPLO radial cut technique following TTA Rapid cage removal in the case of persistent lameness after initial surgery.

## 2. Materials and Methods

### 2.1. Patient

A six-year-old male Labrador retriever (38 kg) with persistent lameness after TTA Rapid was referred for assessment of right pelvic limb lameness. The dog underwent the initial TTA surgery six months prior to referral due to cranial cruciate rupture. According to the owner, the dog had never regained complete limb function following the initial TTA procedure. During orthopedic examination, the lameness was subjectively graded as 3/5 and the positive drawer and tibial compression tests were performed.

### 2.2. Measurements

Preoperative craniocaudal and mediolateral radiographs of the stifle joint were obtained under sedation and surgical planning for a TPLO radial cut was undertaken using vPOP-PRO software (ver. 2.9.2). Preoperative radiographs showed complete osteointegration of TTA Rapid cage to the tibia, signs of stifle joint effusion and the presence of osteophytes consistent with osteoarthritis (OA) associated with cranial cruciate disease. The measurements were performed based on previously described methods; the preoperative tibial plateau angle (TPA) was 27° and the patella ligament angle relative to the tibial plateau (PLA) was 102° [16] (Figure 1A). A decision was made to completely remove the TTA Rapid cage and all screws to have enough room to perform a TPLO radial cut.

### 2.3. Surgery

Prior to surgery, the patient received dexmedetomidine (Dexdomitor, Zoetis, Rhodes, NSW, Australia) and butorphanol (Torbugesic, Pfizer Trading, New York, NY, USA) as premedication. General anesthesia was induced with propofol (2–4 mg/kg IV) and maintained with isoflurane in 100% oxygen. Cefazolin (22 mg/kg IV) was administered 30 min prior to surgery and then after 90 min.

The surgical approach was started from the medial parapatellar skin incision. The subcutaneous tissues were dissected blindly and a medial mini parapatellar arthrotomy was performed for initial inspection of the stifle, to examine any meniscal lesions by visualization and probing. The menisci were intact and then not released. A medial approach to the proximal tibia was performed and TTA screws were gently removed. By using a mallet, a small osteotome was advanced in a mediolateral direction against and parallel to the external sides of the TTA Rapid cage to a depth approximately equal to the length of the cage. After the rim of the cage appeared to be separated from the tibia, the cage was rocked back and forth in a proximal-distal direction by using a periosteal elevator until it was evidently loose. Cranio-caudal manipulations were clearly avoided to minimize fracture of the tibial tuberosity. Gentle steady traction was applied to complete the TTA Rapid cage removal. A 24 mm crescentic TPLO saw blade was used. The location of the tibial osteotomy was determined by preoperative planning from a mediolateral radiograph. Due to not having enough room for the TPLO plate and jig, the procedure was performed without a jig. The osteotomy was performed with an oscillating saw (AESCULAP^®^ Acculan 4 TPLO Saw, B Braun, Melsungen, Germany). The tibial plateau segment was rotated to 8 mm and temporarily fixed in a position with a 1.6 mm Kirschner pin. The 3.5 mm anatomically pre-countered TPLO plate (Iwet, Grabowka, Poland) was positioned on the caudal part of the proximal tibia in a way that best fit the bone surface and osteotomy. The TPLO plate was attached to the tibia using two 3.5 mm standard cortical screws for dynamic compression and five 3.5 mm locking screws (Figure 1B). The gap left after removing the TTA cage was filled with a resorbable bone substitute (PerOssal^®^, OSARTIS GmbH, Münster, Germany). A 1.6 mm Kirschner pin in the cranio-caudal direction was inserted for additional stabilization. The incision was closed in three layers. Two orthogonal postoperative radiographs were performed to evaluate the implants’ position, apposition of the bone fragments and postoperative TPA. Postoperative radiographic TPA was 10°.

### 2.4. Follow-Up

A lameness score evaluation, client satisfaction and radiographic follow-up were performed at 4 weeks, 8 weeks and 6 months postoperatively. Stifle radiographs were repeated under sedation using medetomidine (25 μg/kg IM). All received images were evaluated by the Authors. Radiographs were critically assessed for implant position, alignment and bone healing of the osteotomy and TTA gap site. No postoperative complications related to the TPLO procedure or surgical site infection (SSI) occurred.

## 3. Results

At the 4-week follow-up, the dog significantly improved after surgery with a lower subjective lameness score of 1/5 at a walk and at a trot. Kirschner pins and TPLO plate were aligned and positioned without displacement. Radiographically, the disappearance of the TTA gap and osteotomy gap and the presence of bridging callus were noted (Figure 1C). The owner was already satisfied with weight bearing on the limb at this stage after the procedure. At the 8-week follow-up, the dog used the limb without any lameness (0/5). The TPLO implant and Kirschner wires were still aligned and positioned without displacement compared to immediately postoperative radiography. Radiologically, excellent healing with fusion of the gap and disappearance of the osteotomy line was seen. Six months postoperatively, no lameness was detected at a walk and at a trot. The owner was completely satisfied and reported the dog being free from lameness even after long walks. Radiographic evaluation showed complete bone healing with disappearance of the gap, with the TPLO plate and Kirschner pins in place (Figure 1D). Despite complete radiographic healing and excellent clinical outcome, 6 months postoperative radiographs showed progressive signs of osteophytosis consistent with osteoarthritis.

## 4. Discussion

To the best of our knowledge, this is the only case that shows the viability of applying a TPLO radial cut to address chronic lameness and instability after TTA Rapid cage removal.

Prolonged knee joint instability has been recognized as a potential consequence following proximal tibial osteotomies. In comparison to TPLO procedures, the reported score of chronic stifle instability appears to be greater following TTA procedures [15,16]. The relationship between prolonged instability and poor limb function has yet to be established. However, chronic instability can contribute to the advancement of osteoarthritis, late meniscal tears and the “pivot shift” phenomena [5,17,18]. The greater prevalence of knee instability following TTA has been hypothesized as a risk factor for the reported increased incidence of late meniscal tears compared to TPLO [17,18]. Several mistakes in surgical planning also have the potential consequences of under- or overestimation of the required TTA or selection of an inappropriate cage size [19]. It is well reported that an effective TTA procedure relies on a balance between the extensor and flexor muscles that span the stifle joint. The dissection of some important muscles during the surgical approach potentially could reduce the stabilizing force exerted by these muscle–tendon units [19].

The reason for persistent instability after TTA surgery is still unknown; however, there are two proposed theories including faults in surgical planning and execution of the procedure or in the biomechanical concept [19]. The TTA concept relies on the advancement of the tibial tuberosity such that the forces acting on the tibial plateau during the mid-stance phase of the gait, when the stifle is at 135°, become perpendicular to the straight patella ligament. Inadequate advancement of the tibial tuberosity may result in an angle greater than 90° between the tibial plateau and the patella ligament during the mid-stance phase of locomotion, which may be responsible for chronic positive cranio-caudal tibial thrust [20]. To achieve appropriate tibial tuberosity advancement, precise preoperative planning is mandatory. Multiple radiographic methods of TTA planning have been proposed, all correlated to stifle flexion angle [21]. There is a lack of reports in the literature comparing the variation of the stifle flexion angle in standing dogs of different breeds, especially for chondrodystrophic breeds and small-breed dogs. It is expected that the 135° angle is not acceptable for TTA planning in any dog [21].

One of the causes of lameness after stifle surgery could be late meniscal injury. Some clinical studies reported that the severity and frequency of eventual meniscal pathology is higher after TTA compared to TPLO at short-term and long-term evaluations [22]. In an in vivo experimental study by Engdahl et al. (2021), 5 of 15 (33%) stifle joints had persistent cranial tibial thrust following TTA surgery [23]. Another in vivo clinical study reported by Schwede et al. detected persistent cranial tibial subluxation in 100% of TTA-treated knees using fluoroscopic evaluations [15]. However, there were no differences in the lameness scores between limbs in which a TPLO and TTA were performed [24].

Mobility of bone fragments at the osteotomy line, even without significant loss of reduction, has also been implicated in increased rates of complications, especially in large- and giant-breed dogs [25]. Several studies reported an increased rate of post-operative TPLO complications in large-breed dogs using traditional non-locking plates [25]. Locking and anatomically contoured TPLO plates are now freely commercially available through plenty of companies, with screw sizes ranging from 1.5 mm to 4.5 mm [26]. The precise execution of the TPLO has significantly increased procedural efficiency and minimized clinical complications. Computer software is now widely used in veterinary specialty clinics to design TPLO osteotomies by templating the size of the radial saw blade and establishing reference points to optimally center the osteotomy on the tibial intercondylar eminence [26]. The general rule of thumb is to preserve the tibial tuberosity width of not less than 1 cm in large- and giant-breed dogs and to keep the narrowest part of the tuberosity away from the insertion of the patellar ligament [24,26]. Even anti-rotational Kirschner wire placement has been investigated, with recommendations for placement proximal to the patellar ligament insertion to decrease the complication risk [24]. As a result of this increased understanding of surgical techniques and execution, along with advancing implant technology, the TPLO is now one of the safest and most reproducible elective orthopedic operations available [26].

The preoperative planning of TPLO does not require a stifle flexion angle of 135° [20]. The tibial plateau angle was measured using a lateral radiographic projection of the affected limb [5,27,28]. The aim of TPLO is to achieve a postoperative TPA of 5–6° and thereby completely eliminate cranial tibial thrust while walking [5,27,28]. In this case report the postoperative TPA was 10° which is a little underrotated. The main reason for this is the challenges with the previous TTA cage removal and with the fitting of the TPLO plate to the bone. There are several reports where an interobserver TPA measuring variation ranged from 0.8° to 4.8° and intraobserver variability was 1.5–3.4°. The surgeons chose TPLO surgery for the revision after TTA due to the preoperative planning and measurements of the proximal tibia. Serrani et al. [29] reported the performance of the TPLO after TTA without removing the TTA rapid cage. The TPLO plate was positioned just caudal to the TTA cage. The Authors emphasize that this is an essential preoperative precaution since detaching an osteo-integrated TTA cage is difficult and increases surgical time as well as the probability of iatrogenic tibial fracture. Leaving the TTA implants in place during revision surgery, in our opinion, increases the possibility of surgical site infections [23].

This report has several limitations including that it is a case report. A kinematic study of the dogs prior to and following surgery might provide a more reliable pre- and post-operative assessment of the lameness. To confirm the suitability of the TPLO procedure after TTA Rapid cage removal, a similar procedure must be performed on a larger number of patients.

## 5. Conclusions

In conclusion, TPLO surgery was successfully performed after TTA Rapid cage removal to treat persistent stifle joint instability and chronic lameness in a six-year-old Labrador retriever. TPLO following initial TTA yielded good to excellent clinical outcomes and improved clinical function. In our case report, TPLO was performed after the TTA implant was removed in order to improve stifle stability and function.

## Figures and Tables

**Figure 1 animals-13-03444-f001:**
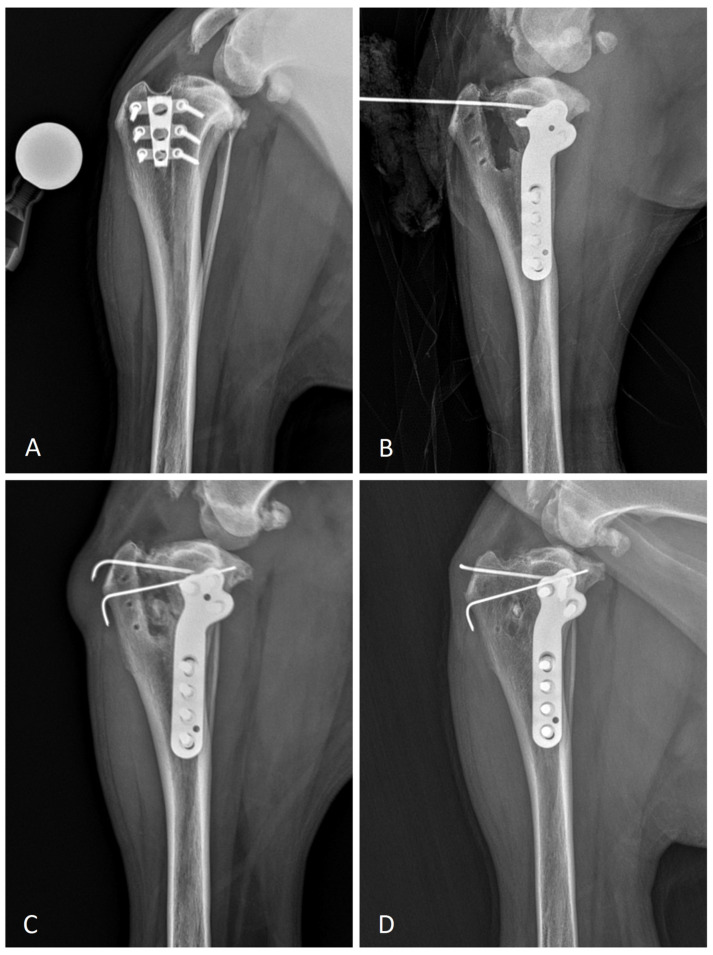
(**A**) Mediolateral radiograph of the stifle joint six months after initial TTA Rapid surgery; (**B**) intra-operative radiograph after TTA Rapid cage was removed. Anatomically pre-countered TPLO plate was positioned on the caudal part of the proximal tibia in a way that best fit the bone surface and osteotomy; (**C**) 8 weeks postoperative mediolateral radiograph. The disappearance of the TTA gap and osteotomy gap and the presence of bridging callus were noted; (**D**) 6 months postoperative mediolateral radiograph. Complete bone healing with disappearance of the gap and the TPLO plate and Kirschner pins in place was noted.

## Data Availability

Data and materials are available from the corresponding author upon reasonable request.

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
