# Peer review of "Tibial Plateau Leveling Osteotomy following Tibial Tuberosity Advancement Cage Removal: A Case Report"

_animals, 2023, doi:10.3390/ani13223444_

Round 1

Reviewer 1 Report

Comments and Suggestions for Authors

Tibial plateau leveling osteotomy following tibial tuberosity 2 advancement cage removal: a case report.

Review:

Thank you very much for providing this interesting case report. This is a great road map for similar cases. The reviewer thanks the authors for this contribution. Most of the comments are regarding language and translational suggestions.

Line 11:  The reviewer recommend to strike ……”and TPLO techniques causing chronic lameness.”  Since this article is about a failed TTA.

Line 12 : Strike initial

Line 13: Please remove “to” in graded to 3/5 …

Line 16: Please add “The “ decision

Line 22: Please change sentence to “a tibial plateau leveling osteotomy (TPLO) in a dog with joint instability and persistent lameness following a tibial tuberosity advancement surgery (TTA) Rapid. “

Line 24: Please change sentence to : A six-year-old male Labrador retriever (38kg) with tibial plateau angle of 27° and a patella ligament to tibial plateau angle of 102° and persistent lameness lasting six month after TTA Rapid surgery has been examined.

Line 47: However, the excellence of one surgical technique over another has not been settled in terms  - please change to: However, it is unclear which method is superior in regards to preventing pain and progression of osteoarthritis.

Line 64: right not “wright”

Line 83: does this mean the meniscus was intact ? We do not really say untouched unless the authors mean to say they did not probe the meniscus, which I think they do not mean to say. Was the meniscus  then released or not?

Line 91: Was a jig used for this surgery to secure the tibial plateau and to optimize the rotation?

Line 77: A decision was made

Line 101: Strike Supplementary another 1.6 mm – please replace with “A 1.6 mm…”

Line 105: The post op TPA was 100 which is a bit underrotated. One assumes this is due to the challenges with the previous TTA removal. That could be discussed a bit more in the discussion part of this case report.

Line 111: replace “od” with or

Line 112 : strike were

Line 155: The reviewer suggests to add “could” reduce the stabilizing force

Line 187: six-year old Labrador retriever

Comments on the Quality of English Language

I have added some editing suggestions above in the previous paragraph. 

Author Response

Dear Reviewer,

Thank you for your constructive and insightful review.

  1. We removed word “TPLO” at Line 11 as you suggested. We agree it is report about a failed TTA.
  2. We deleted “initial” at Line 12.
  3. We deleted “to” in graded to 3/5at Line 13.
  4. We added at Line 16 “The“ decision.
  5. Line 22 we have changed sentence as You suggested.
  6. Line 24: we have changed sentence as You suggested.
  7. Line 47: we have changed sentence as You suggested.
  8. Line 64 : we have changed wright to “right”.
  9. Line 83: the Authors mean that meniscus was intact. Thank you for your clarification.
  10. Line 91: It was jigless procedure due to the small room in proximal tibia for the TPLO plate and Jig.
  11. Line 77 we add “A”
  12. Line 101 we replace sentence as you suggested.
  13. Line 105 we add some discussion about postoperative radiographic TPA that was about 10 degree.
  14. Line 111 we have corrected “od” to “or”
  15. Line 112 we have removed “were”
  16. Line 155 Authors added  “could”
  17. Line 187: we have made changes in sentence “six-year old Labrador retriever”

One more time thank you for your report.

Reviewer 2 Report

Comments and Suggestions for Authors

This is an interesting case report presentation.

Line 55-56: Needs to be completely restructured. (To the authors’ knowledge, the use of TPLO radial cut surgery is unreported procedure to treat persistent lameness and instability after TTA Rapid without implant removal)

Do the authors believe that the previous surgical technique, the TTA rapid caused instability due to a fault in surgical planning or technique? 

Since the difference from Serrani's case series and this case report is the removal of the TTA cage, could the authors provide scientific research to support the statement at line 178-179 ( leaving the TTA implants during the revision surgery may pose a risk of surgical 178 site infections)?

Comments on the Quality of English Language

The writing of the article needs much improvement. I suggest some proofreading tool.

Author Response

Dear Reviewer,

Thank you for your constructive and insightful review.

  1. We reconstructed completely the Line 55-56: 
  2. We do not have preoperative radiographs (pre-TTA) because the dog was referred to our clinic. But in our opinion Figure 1A shows that preoperative TPA slope was to high for TTA surgery.
  3.  We have provided scientific research to support the statement at line 178-179. Engdahl et al 2021  Risk factors for severe postoperative complications in dogs with cranial cruciate ligament disease – A survival analysis, “Surgical site infections or complications related to the surgical implants were the most common severe post- operative complications (Table 4).”  Our intension of this statement was leaving any foreign body (implant, plate, screw) potentially increases the risk of complications due to the biofilm formed on the implants.

One more time thank you for the review.

Reviewer 3 Report

Comments and Suggestions for Authors
  • A brief summary (one short paragraph) outlining the aim of the paper, its main contributions and strengths.

This paper describes a case report- a revisional TPLO surgery of a Labrador retriever after the initial TTA Rapid followed by persistent lameness and intability. Although TPLO revisional surgeries after inital TTA treatment have been reported, thus far, explanting a previously implanted TTA cage before performing a TPLO has not reported.

  • General concept comments
    Article: This article is a case report. It would be better if this was presented as a case series, although the specifics of this case and the lack of cases for a case-series is understandable. TTA and TPLO are competing surgeries, and normally are seldom performed in the same patient. Cases of both surgeries in the same pateient do exist, but the implants of previous surgery in those reports are left in situ. This case report describes an alternative procedure, where a fully-integrated TTA cage is explanted prior to a TPLO procedure. This can be of value to v eterinary orthopedic specialists as there are major complication concerns with this type of procedure, and it has not yet been reported.

Review:

·         This case report successfully reviews the relevant topic before describing the case. The possible gap in presentation identified and not referenced is (Jeong J, Jeong SM, Kim SE, Lewis DD, Lee H. Subsequent meniscal tears following tibial tuberosity advancement and tibial plateau leveling osteotomy in dogs with cranial cruciate ligament deficiency: An in vivo experimental study. Vet Surg. 2021 Jul;50(5):966-974. doi: 10.1111/vsu.13648. Epub 2021 Apr 29. PMID: 33928658.).

The authors are encouraged to include this reference into the introduction, as it might slightly change the overall tone of the introduction and the results of this differ from the results and conclusion of other cited papers (8-10, 13).

·         The figures are clear and non-surplus. They provide a good understanding of the described procedure and present a good mid-term follow-up.

  • The manuscript Is clear, relevant for the field and presented in a well-structured manner. The procedure described is of interest to veterinary orthopedic specialists.
  • The cited references are mostly recent publications and relevant. There are no self-citations.
  • The manuscript’s results are reproducible based on the details given in the methods section.
  • The conclusions are consistent with the evidence and presented arguments.
  • The ethics statements and data availability statements are not necessary.

I thank the editor(s) for the opportunity to review this article.

Best regards.

Comments on the Quality of English Language

Minor editing of English language required

Author Response

Dear Reviewer,

Thank you for your constructive and insightful review. 

We have made corrections to the introduction and added suggested publication. Thank you very much for your suggestions because it is a very interesting article. Thank you again for reviewing our case report.

Round 2

Reviewer 1 Report

Comments and Suggestions for Authors

Comments on the Quality of English Language

Author Response

Dear Reviewer,

thank you for your thorough report.

We have corrected all English translation as you suggested (starting from Line 11 until Line 236). We believe that this will improve the quality of the publication. 

Thank you.

Best regards

Reviewer 2 Report

Comments and Suggestions for Authors

Please explain this sentence in your article

'To the best of our knowledge, this is the first report documenting the feasibility of TPLO 141 radial cut to treat the persistent lameness and instability following TTA Rapid cage removal'

considering this sentence from Serrani et al., 2022 (Tibial Plateau Leveling Following Tibial Tuberosity Advancement: A Case Series) 'In Cases 4 and 6, the original implants were removed' 

Thank you!

Author Response

Dear Reviewer,

thank you for your report. In publication of Serrani et al., 2022 in Case 4 and 6 there were cases where MMP (Modified Maquet Procedure) procedure were performed. The MMP procedure uses a titanium foam wedge without screws to advance the front of the tibia forward. In our case report we have removed TTA Rapid cage and 6 screws which is firstly reported.

Thank you one more time.

Regards